# All-optical control of phase singularities using strong light-matter coupling

Philip A. Thomas [1✉], Kishan S. Menghrajani[1] & William L. Barnes [1✉]

Strong light-matter coupling occurs when the rate of energy exchange between an electromagnetic mode and a molecular ensemble exceeds competing dissipative processes. The study of strong coupling has been motivated by applications such as lasing and the modification of chemical processes. Here we show that strong coupling can be used to create phase singularities. Many nanophotonic structures have been designed to generate phase singularities for use in sensing and optoelectronics. We utilise the concept of cavity-free strong coupling, where electromagnetic modes sustained by a material are strong enough to strongly couple to the material's own molecular resonance, to create phase singularities in a simple thin film of organic molecules. We show that the use of photochromic molecules allows for all-optical control of phase singularities. Our results suggest what we believe to be both a new application for strong light-matter coupling and a new, simplified, more versatile means of manipulating phase singularities.

[1] Department of Physics and Astronomy, University of Exeter, Exeter EX4 4QL, UK. ✉email: p.thomas2@exeter.ac.uk; w.l.barnes@exeter.ac.uk

Phase singularities occur when the amplitude of light reflected by an object is zero and its phase becomes undefined[1,2]. Singular optics, the field of optics concerned with the exploitation of phase singularities[3], has primarily studied optical vortices in electromagnetic field profiles in real space[4]. Phase singularities can also be observed in photonic dispersion plots, where a response function (such as phase change on reflection) is plotted in parameter space (e.g. as a function of energy and incident angle[5] or geometric parameter[6]). These phase singularities, which lead to sharp phase jumps in reflection spectra[7], could find application in sensing[8–11], potentially enabling a sensitivity three orders of magnitude greater than commercial amplitude-based sensing technology[12,13], and in flat optics, where it is otherwise difficult to induce strong optical phase variations in materials with deeply subwavelength thicknesses[4,14]. Phase singularities have been observed using the Brewster angle[15], surface plasmon resonances[7–9,13], plasmonic lattices[5,12,16], transition metal dichalcogenides[14], optical Tamm states[10] and Fabry-Pérot microcavities[6,11]. The realisation of phase singularities requires careful design of complicated structures using lithography[5,6,12], self-assembly[16] or multilayers[10,11,13]. In these structures, phase singularities can only be observed under a very specific set of conditions, such as one particular incident angle of light.

In this article, we show that phase singularities can be created using strong light-matter coupling. An ensemble of molecular resonators and a confined electromagnetic field can become coupled if the matter and light modes can be excited under the same conditions[17,18]. When the coupling strength exceeds losses to the environment the light and matter modes enter the strong coupling regime, forming hybrid states known as polaritons[19]. Strong coupling could find use in applications such as lasing[20] or the modification of chemical processes[21]; to the best of our knowledge, the generation of phase singularities using strong coupling has not previously been reported. While most strong coupling experiments rely on external structures (such as planar microcavities[17,22] or plasmonic nanostructures[19]) to generate confined electromagnetic fields, our results build on recent work which shows that such structures are not always needed[23–26]. We use such a cavity-free design here to observe phase singularities associated with each newly created polariton state. We use a photochemical reaction to control the number of coupled molecules, and hence light-matter coupling strength, by simple irradiation of light[22]. This allows one to tune into and detune from the phase singularities associated with each polariton branch, modifying the phase sensitivity of the films. Our results demonstrate what we believe to be the first application of strong light-matter coupling to the creation of phase singularities in dispersion plots in a remarkably simple structure.

## Results

**Sample design and characterisation.** We studied silicon substrates coated with a thin film of organic molecules (see diagram in Fig. 1a and film fabrication details in Methods). We used the photochromic molecule spiropyran (SPI), which is transparent to visible light and can be converted to merocyanine (MC) by exposure to ultraviolet light. MC has a strong absorption peak at energy $E = E_{MC} = 2.22$ eV (see Supplementary Information for SPI/MC molecular structures and MC transmission spectrum) and can be

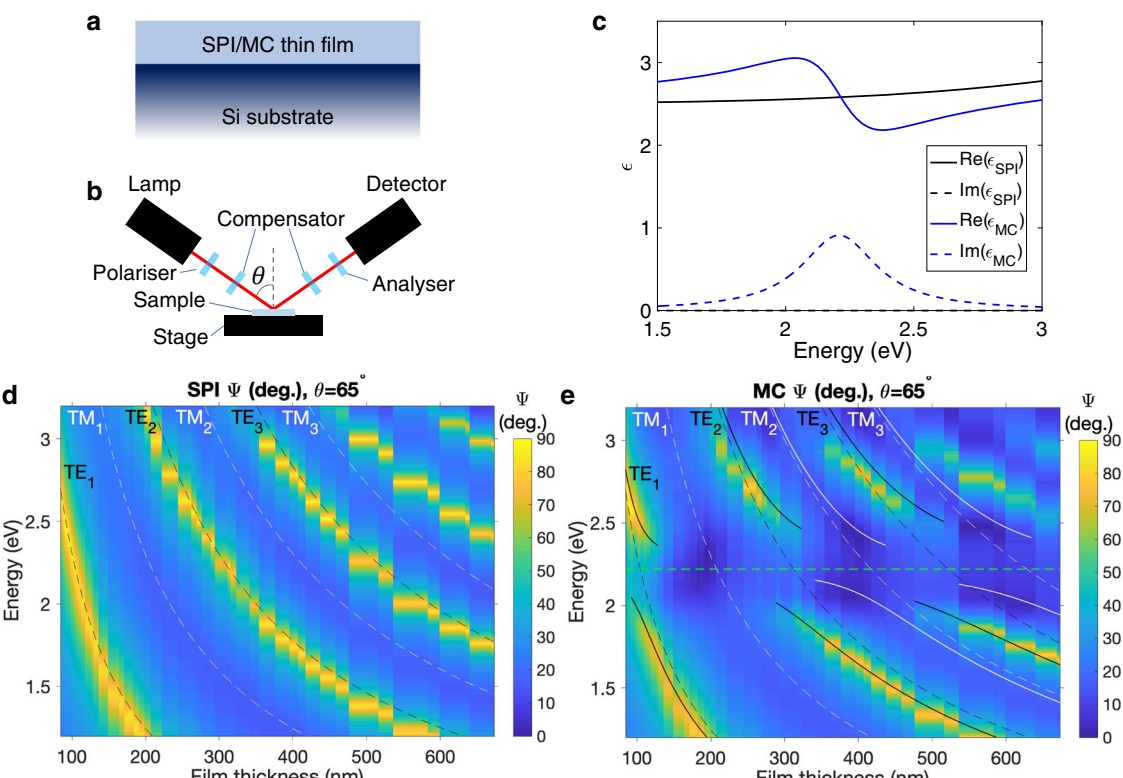

**Fig. 1 Cavity-free strong light-matter coupling. a** Sample design: SPI/MC film on silicon substrate. **b** Ellipsometer schematic. **c** Complex permittivities $\epsilon$ of SPI and MC, as derived from ellipsometry for a film of thickness 109 nm. **d, e** Dispersion plots constructed using the ellipsometric parameter Ψ for **d** SPI and **e** MC films over a range of thicknesses at fixed angle $\theta = 65°$. The dashed lines in **d, e** indicate the positions of the uncoupled TE (black) and TM (white) leaky modes. The green dashed line in **e** at $E = 2.22$ eV indicates the position of the MC molecular resonance. The solid lines show the predicted positions of polariton branches using the 2N coupled oscillator model. The coupled TE polariton branches (black) were fit with a coupling strength $g = 225$ meV. The TM$_2$ and TM$_3$ polariton branches (white) were fit with $g = 185$ meV and $g = 200$ meV, respectively. We did not perform a coupled oscillator fit for the TM$_1$ mode since it does not show any clear anticrossing.

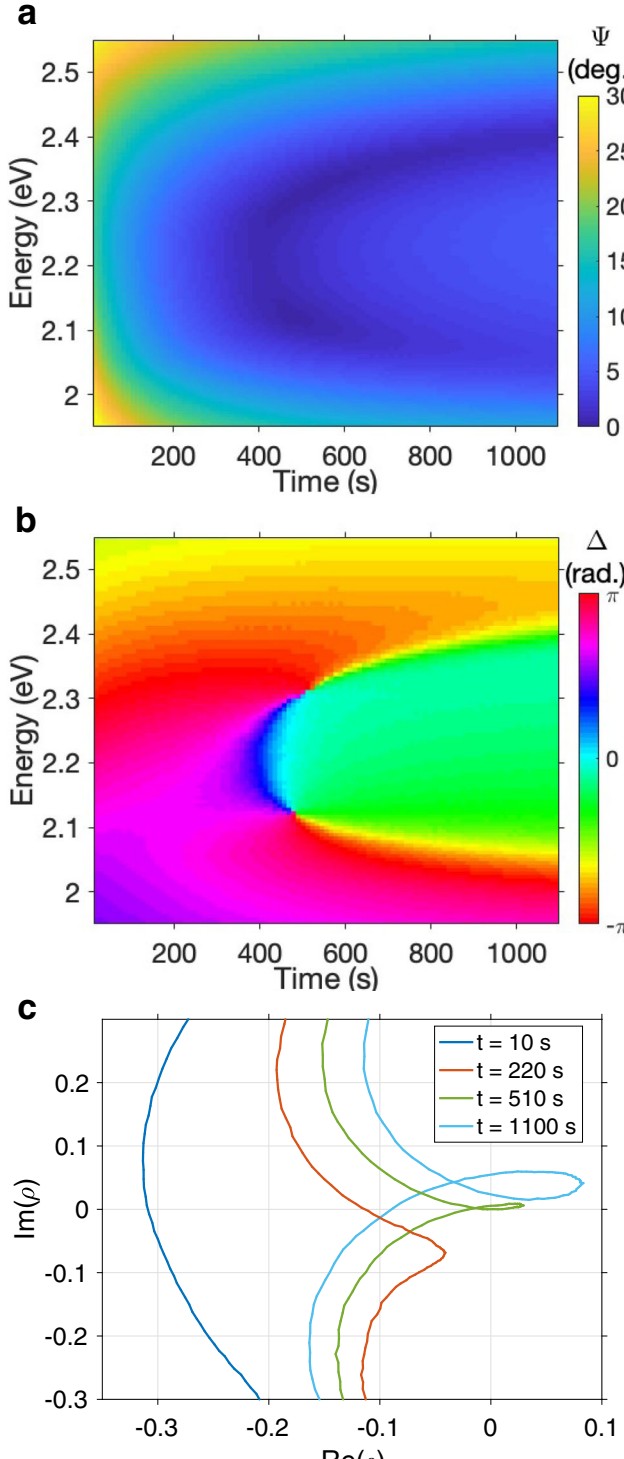

**Fig. 2 Creation of phase singularities in an SPI/MC thin film.** The ellipsometric parameters (**a**) $\Psi$, (**b**) $\Delta$ and (**c**) $\rho$ plotted for a thin SPI film (thickness $L = 407$ nm, measured at an incident angle $\theta = 65°$) while exposed to UV radiation. As UV exposure time increases, SPI undergoes photoisomerisation to MC and the system transitions from the weak coupling regime to the strong coupling regime.

reconverted back into SPI after exposure to visible radiation[27]. Reversible SPI/MC photoisomerisation has been exploited for all-optical control of strong coupling in Fabry-Pérot microcavities[22] and lasing in plasmonic lattices[28]. The permittivites of SPI and MC

(determined using ellipsometry—see Methods) are plotted in Fig. 1c.

We characterised our samples using spectroscopic ellipsometry (schematic in Fig. 1b; see also Methods). Ellipsometry measures the ratio of the p- and s-polarised reflection coefficients, $\rho$, expressed as $\Psi$ and $\Delta$[29]:

$$\rho = \frac{r_p}{r_s} = \tan(\Psi)e^{i\Delta}. \qquad (1)$$

$r_p$ and $r_s$ are respectively the p- and s-polarised Fresnel amplitude reflection coefficients. The amplitude of $\rho$ ($\tan(\Psi)$) is the ratio of the moduli of $r_p$ and $r_s$; the phase of $\rho$ ($\Delta$) gives the difference between the phase shifts experienced by p- and s-polarised light on reflection. Our ellipsometer's xenon light source emits low-intensity ultraviolet radiation, allowing us to monitor the conversion of SPI to MC.

The $\Psi$ spectra for SPI films with a range of thicknesses $L$ (84 nm $< L <$ 680 nm) measured at an incident angle $\theta = 65°$ are plotted in Fig. 1d to create a dispersion plot. The large impedance mismatches at the Si/SPI and SPI/air interfaces give rise to a series of transverse electric (TE) and transverse magnetic (TM) leaky modes (sometimes called quasi-normal modes[30]). For TE modes, $r_s < r_p$, giving maxima in $\Psi$; the positions of uncoupled TE leaky modes are indicated in Fig. 1d, e by dashed black lines. For TM modes, $r_p < r_s$, giving minima in $\Psi$; the positions of uncoupled TM leaky modes are indicated in Fig. 1d, e by dashed white lines. In Fig. 1e we plot $\Psi$ spectra for the films in Fig. 1d after photoconversion to MC. Leaky modes can strongly couple to MC's molecular resonance. There is clear anticrossing of the TE leaky modes around $E = E_{MC}$[26], a signature of strong coupling between the TE leaky modes and MC resonance[19]. TM modes are generally weaker than TE modes since $r_p$ is much lower than $r_s$ at the SPI/MC-Si interface[31] at $\theta = 65°$. A coupled oscillator fit (Methods) suggests that the TM$_2$ and TM$_3$ modes have coupling strengths of $g_{TM2} = 185$ meV and $g_{TM3} = 200$ meV, meaning that they fulfil the strong coupling resolution criterion[19]:

$$g_{TM2,3} > \frac{1}{4}(\gamma_{MC} + \gamma_{TM2,3}), \qquad (2)$$

where $\gamma_{MC} \approx 360$ meV is the MC resonance linewidth and $\gamma_{TM2} \approx 350$ meV and $\gamma_{TM3} \approx 240$ meV are the TM$_{2,3}$ leaky mode linewidths, respectively. The TM$_1$ mode (for which $\gamma_{TM1} \approx 430$ meV) does not fulfil this criterion and there is no clear anticrossing of the TM$_1$ mode in Fig. 1e.

**Observation of phase singularities.** In Fig. 2 we show how the ellipsometric parameters $\Psi$, $\Delta$ and $\rho$ change as a thin SPI film is converted to MC. For this thickness of film ($L = 407$ nm) and incident angle of light ($\theta = 65°$) the energy of the TM$_2$ leaky mode matches that of the MC resonance. Fig. 2a shows how $\Psi$ changes with increasing MC concentration. After 400 s of UV exposure two distinct modes appear; as the MC concentration increases, the mode splitting increases. This behaviour is consistent with the transition from the weak to strong coupling regime[32]. The minimum value of $\Psi$ drops dramatically in the strong coupling regime, with one point on each branch going below $0.1°$ (comparable with experimental error). In Fig. 2b we plot $\Delta$, which shows the same phase behaviour between the two polariton branches that we have previously interpreted as a signature of strong coupling[32]. Shortly after entering the strong coupling regime, we observe two phase singularities: the first, at $t = 470$ s, on the lower polariton branch; the second, at $t = 510$ s, on the upper polariton branch. They correspond to positions on the upper and lower polariton branches in Fig. 2a at which $\Psi < 0.1°$. When a loop is traced around a phase singularity in two-dimensional parameter space a phase of

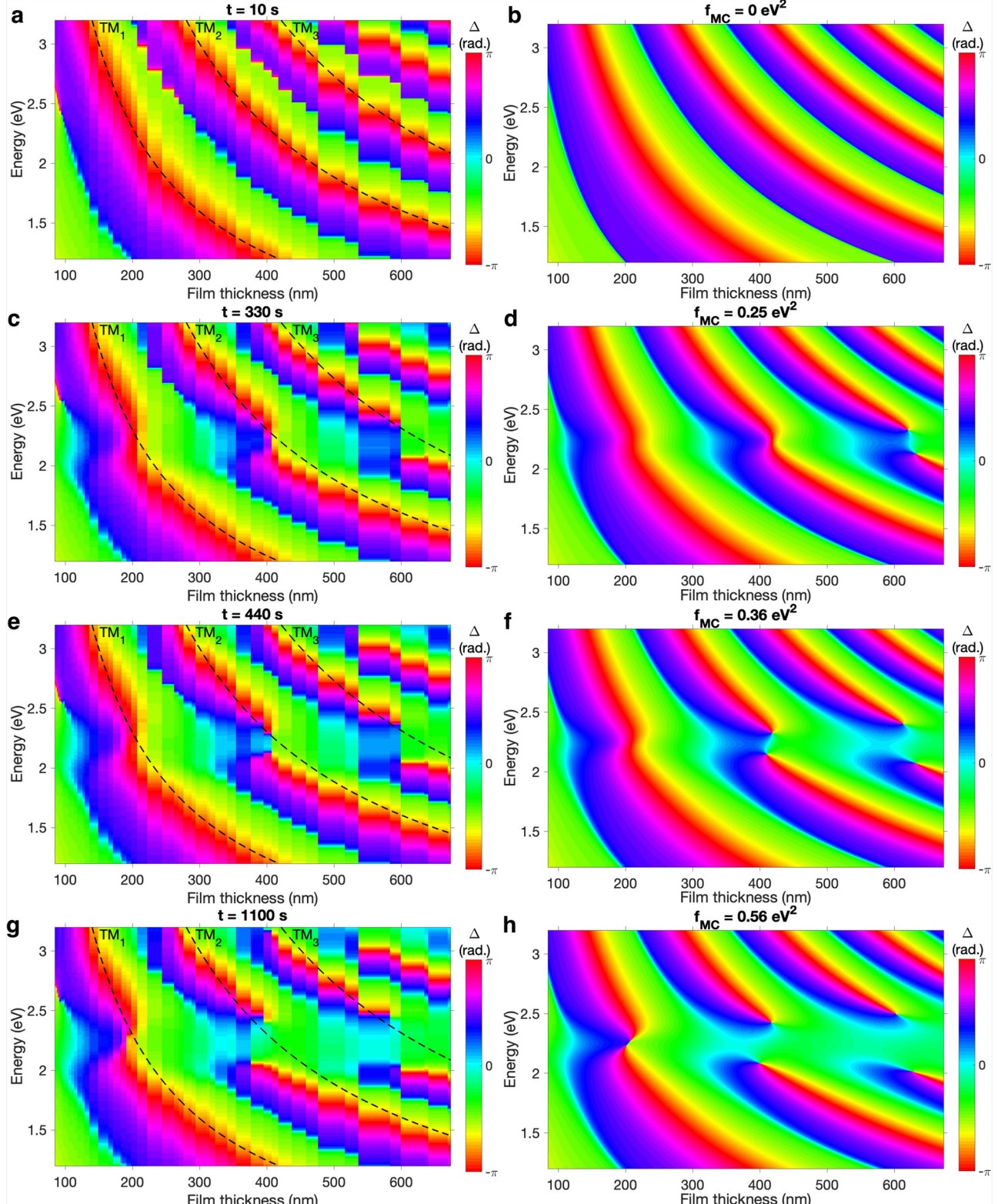

**Fig. 3 Creation of phase singularities for a range of film thicknesses. a, c, e, g** Experimental and **b, d, f, h** calculated (Fresnel approach) dispersion plots constructed using the ellipsometric parameter Δ for SPI/MC films over a range of thicknesses at fixed angle $\theta = 65°$. The MC concentrations were varied in experimental plots by UV exposure time (**a** $t = 10$ s, **c** $t = 330$ s, **e** $t = 440$ s, **g** $t = 1100$ s) and in calculated plots by varying the Lorentz oscillator strength of MC (**b** $f_{MC} = 0$ eV², **d** $f = 0.25$ eV², **f** $f = 0.36$ eV² and **h** $f = 0.56$ eV²). The positions of the uncoupled TM modes are indicated by the black dashed lines in **a, c, e, g**.

$2\pi C$ is accumulated, where $C$ is the topological charge associated with each phase singularity. The phase singularities in Fig. 2b have topological charges of $C = +1$ and $C = -1$, indicating that the total topological charge of the system is preserved[6,14].

Figure 2c shows $\rho$ at four times: $t = 10$ s, when the film is almost entirely composed of SPI molecules; $t = 220$ s, when some SPI has undergone conversion to MC, but the MC density in the film remains too low for strong coupling; $t = 510$ s, coincident with the upper polariton phase singularity, when the MC density in the film is high enough to place the system within the strong coupling regime; and $t = 1100$ s, the final measurement when most SPI has been converted into MC. The evolution of $\rho$ as the system enters the strong coupling regime resembles what we have observed in other strongly coupled systems[32]: the initial TM$_2$ leaky mode is represented by an arc; in the weak coupling regime this arc is perturbed by the presence of MC; in the strong coupling regime a new loop is observed. As the MC concentration increases, the loop both increases in size and touches the point $\rho = 0$ for $t = 510$ s (green loop) and $t = 470$ s (not plotted for clarity). Provided the strong coupling loop passes over $\rho = 0$ as the system evolves, we are guaranteed to observe two phase singularities.

In Fig. 3 we plot $\Delta$ for SPI/MC films with thicknesses $84$ nm $< L < 680$ nm. The equivalent $\Psi$ data is plotted in Supplementary Fig. S2. Each film was exposed to UV radiation under the same conditions (with $\theta = 65°$). We plot the phase response after (a) $t = 10$ s, (c) $t = 330$ s, (e) $t = 440$ s and (g) $t = 1,100$ s of UV irradiation. The dashed black lines indicate the positions of the uncoupled TM modes in the SPI films. These results are well reproduced in Fresnel calculations (Fig. 3b, d, f, h) by varying the Lorentz oscillator strength of the MC resonance. Initially, in Fig. 3a, b, we observe no phase singularities associated with any TM leaky modes. We first observe the splitting of the TM$_3$ leaky mode (the best-confined of the plotted TM modes) in Fig. 3c, d, followed by splitting of the TM$_2$ mode in Fig. 3e, f. The topological charges of the phase singularities associated with each anticrossing are $C = +1$ and $C = -1$. Therefore, as in Fig. 2b, the topological charge is preserved as each pair of phase singularities is created. We did not observe clear anticrossing of the TM$_1$ mode in Fig. 1e; likewise, in Fig. 3g, h we do not unambiguously observe the creation of phase singularities associated with the TM$_1$ mode. Phase singularities can also be created by coupling to TE modes under slightly different conditions; see Supplementary Section S3.

## Discussion

Exciton-polariton condensates arising from strong light-matter coupling have been used to explore a wide range of phenomena[33], including the manipulation of real space phase singularities[34–37]. These phase singularities (and their associated optical vortices) are typically observed in electromagnetic field profiles and located at particular spatial coordinates[4]. The phase singularities in Fig. 3 do not occur in physical space but in a response function in parameter space: they can be observed in dispersion plots but not in real-space plots. They are perhaps most similar to the work of Berkhout and Koenderink[6], where phase singularities in plasmon antenna array etalons were observed by plotting reflectivity and phase spectra over a range of etalon spacings. It is also possible to identify phase singularities by plotting reflectivity and phase spectra for a range of incident angles[5,11,14]. We have shown that it is possible to manipulate phase singularities in photonic dispersion plots in the time domain over the course of seconds, similar to previous observations of phase singularities in the spatiotemporal domain over the course of femtoseconds[38–41]. The qualitative similarities

between our results and those in the spatiotemporal domain highlight the universal nature of phase singularities. Note that we have only studied the forward photochemical reaction (SPI to MC), but since this process can be reversed under exposure to high-intensity visible light[22,28], our system in principle allows one to both create and annihilate phase singularities in dispersion plots.

In conclusion, we have shown that molecules in a thin film can undergo strong coupling to leaky modes in the same thin film of which they are a part, and that entering the strong coupling regime creates pairs of phase singularities. The phase sensitivity of a thin film of a particular thickness can be varied by modifying the incident angle of light or the concentration of molecules in the film. The use of photochromic molecules allowed us to demonstrate all-optical control of phase singularities. Our results highlight what we believe to be both a new application of strong coupling and a new, extraordinarily simple platform for singular optics.

## Methods

**SPI/MC film fabrication.** Polymethyl methacrylate (PMMA, molar weight 450,000) was used as a host matrix for SPI (Sigma-Aldrich; see Supplementary Section 1 for SPI and MC molecular structures and MC transmission spectrum). PMMA was dissolved in toluene. SPI was then dissolved in the PMMA-toluene solution with a weight ratio of 3:2 SPI to PMMA. SPI/PMMA films were deposited on a silicon wafer by spin-coating three layers each with a spin speed of 2000 rpm. This produced film thicknesses in the range 84–681 nm.

**Optical constants of SPI and MC.** Optical constants of SPI and MC were determined using CompleteEASE® 6.51 (J.A. Woollam Co., Inc.). The permittivity of SPI ($\epsilon_{\text{SPI}}$) was modelled as a Cauchy dielectric:

$$\epsilon_{\text{SPI}} = \left(A + B\omega^2 + C\omega^4\right)^2 \tag{3}$$

where $A = 1.584$, $B = -4.171 \times 10^{-34}$ rad$^{-2}$ s$^2$ and $C = 2.105 \times 10^{-64}$ rad$^{-4}$ s$^4$. The permittivity of MC ($\epsilon_{\text{MC}}$) was modelled with a Lorentz oscillator ($\epsilon_L$) and a pole in the ultraviolet ($\epsilon_{\text{UV}}$):

$$\epsilon_{\text{MC}} = \epsilon_\infty + \epsilon_{\text{UV}} + \epsilon_L \tag{4}$$

$$= \epsilon_\infty + \frac{A_{\text{UV}}}{E_{\text{UV}}^2 - (\hbar\omega)^2} + \frac{f}{E_L^2 - (\hbar\omega)^2 - i\hbar\omega\gamma_L}, \tag{5}$$

where $\epsilon_\infty = 0.839$, $A_{\text{UV}} = 102.133$ eV$^2$, $E_{\text{UV}} = 7.970$ eV, $f = 0.7226$ eV$^2$, $E_L = 2.215$ eV, $\gamma_L = 0.3587$ eV.

**Ellipsometry.** Spectroscopic ellipsometry was performed with a J. A. Woollam Co. M-2000XI which measures $\Psi$ and $\Delta$ in the wavelength range 210–1690 nm, with a wavelength step of 1.5 nm for 210–1000 nm and 3.5 nm for 1000–1690 nm. $\Psi$ is the ratio of the field reflection coefficients for $p$- and $s$-polarised light (the moduli of the complex Fresnel reflection coefficients for $p$- and $s$-polarised light, $r_p$ and $r_s$ respectively); $\Delta$ is the phase difference between these coefficients such that $r_p/r_s = \tan(\Psi)e^{i\Delta}$. Ellipsometry measures the ratio of two quantities, making it a very low-noise, sensitive technique. This makes it possible to confidently identify points at which $\rho = 0$ where we would expect to see phase singularities.

**Coupled oscillator fit.** The absence of mid-polariton bands in Fig. 1e suggests that the strong coupling we observe can be best modelled by a 2N coupled oscillator model[42], meaning the coupling between the MC resonance and each[43] leaky mode can be described by the following equation:

$$\begin{pmatrix} E_{\text{MC}} & g_j \\ g_j & E_j \end{pmatrix} \begin{pmatrix} a_{(\text{L,U})j} \\ b_{(\text{L,U})j} \end{pmatrix} = E_{(\text{L,U})j} \begin{pmatrix} a_{(\text{L,U})j} \\ b_{(\text{L,U})j} \end{pmatrix}. \tag{6}$$

$E_{\text{MC}}$ is the MC resonance energy, $g_j$ is the coupling strength between MC and the $j$th leaky mode, $E_j$ is the energy of the $j$th leaky mode, $E_{(\text{L,U})j}$ are the lower and upper polariton branch energies associated with the $j$th leaky mode and $|a_{(\text{L,U})j}|^2$ and $|b_{(\text{L,U})j}|^2$ are the Hopfield coefficients describing the mixing of the MC resonance with the $j$th leaky mode.

**Reporting Summary.** Further information on research design is available in the Nature Research Reporting Summary linked to this article.

## Data availability

Data in support of our findings are available at: https://doi.org/10.24378/exe.3844.

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

## Acknowledgements

P.A.T. and W.L.B. acknowledge the support of the European Research Council through Project Photmat (ERC-2016-AdG-742222: www.photmat.eu). K.S.M. and W.L.B. acknowledge support from the Leverhulme Trust research grant "Synthetic biological control of quantum optics".

## Author contributions

P.A.T. and K.S.M. designed the experiment. K.S.M. fabricated samples and P.A.T. performed measurements. P.A.T. analysed experimental data. P.A.T. and W.L.B. performed calculations. P.A.T. wrote the manuscript with input from W.L.B. and K.S.M.

## Competing interests

The authors declare no competing interests.
