## [Peer review file · Nature Communications]

REVIEWER COMMENTS

Reviewer #2 (Remarks to the Author):

Manuscript “All-optical control of phase singularities using strong light-matter coupling” by P. A. Thomas, K. S. Menghrajani and W. L. Barnes is an experimental work demonstrating the appearance of “phase singularities” in a new relatively simple cavity-free setting. Authors utilize strong coupling of laser light to a thin film of photochromic molecules and formation of polaritons. Controlling the strength of the coupling by the number of resonant molecules, authors demonstrate the appearance of anticrossings in the polariton branches of the energy vs. film thickness dispersion in Fig. 1. Corresponding phase profiles in Fig. 3 confirm appearance of dislocations with opposite twists. The work is technically sound and represent sufficient novelty of interest to the broad audience of Nature Communications.

My main concern is what appears to be a confusing terminology emerging in related fields, namely use of “phase singularities” outside of the original context introduced by J. F. Nye and M. V. Berry in 1974 [Dislocations in Wave Trains, Proc. R. Soc. London A 336, 165 (1974)]. The later wavefront phase singularities, or optical vortices, as field objects are closely related in their features to “phase dislocations” observed in current manuscript and related works, e.g. quantization of the twist around dislocations, nucleation of vortex-antivortex pairs, etc. Despite their differences they indeed must belong to a general class of optical phenomena reviewed recently by J. Ni et al as “Multidimensional phase singularities in nanophotonics” [Science 374, 418 (2021)], distinguishing singularities in the spatial, spatiotemporal, and momentum domains.

Similarly, the term “singular phase optics” is in direct confusion with “singular optics” of transverse laser modes carrying optical angular momentum, see, e.g.

M. S. Soskin and M. V. Vasnetsov, Prog. Opt. 42, 219 (2001),

A. S. Desyatnikov, Yu. S. Kivshar, and L. Torner, Prog. Opt. 47, 291 (2005),

M. R. Dennis, K. O'Holleran, and M. J. Padgett, Prog. Opt. 53, 293 (2009).

Ironically, the only mentioning of “phase singularity” in Ref. [1] of present manuscript is related to wavefront singularity.

Authors are encouraged to put their findings in a broad context of established field of singular optics and references above, in particular discuss how “phase singularities” in the polariton branches of measured spectra here may relate to the spatiotemporal wave singularities of polaritons and their light component.

Reviewer #3 (Remarks to the Author):

It is quite difficult to understand the purport of this paper. The experimental results which demonstrate the possibility to create phase singularities in optical fields hosted by thin film of molecular materials may be, in principle, appropriate for the publication. However, the publication in Nature Communications implies that the paper should be sufficiently interesting to broader readership of the journal, being clearly related to sufficiently general and clearly defined topics. The present manuscript does not meet this condition. The presentation is strongly focused on details of the experiment, without showing an intention to relate the results to general themes of optics. In particular, a commonly known fact is that phase singularities appear in cores of optical vortices. The present manuscript completely ignores this relation. Such concepts as vortices or vorticity are not even mentioned in the text. Further, it is commonly known that phase singularities are characterized by the topological charge (winding number). If it is S , then the amplitude field around the singularity decays at $r \rightarrow 0$ as r^{-S} (r is the distance from the singularity point). The text twice quickly mentions the topological charge, without specifying values of the charge observed in the experimental pictures, and no results are reported or even briefly mentioned for the amplitude structure, that should be related to the phase singularities.

A paper which is so tightly focused on technicalities of observing phase distributions of optical fields in thin molecular films, without placing it in the general context of optics and physics of vortices, may be appropriate for a strongly specialized journal, such as Applied Optics or, maybe, Optics Express. The paper, in its present form, seems completely inappropriate for an interdisciplinary journal, such as Nature Communications.

Reviewer #4 (Remarks to the Author):

Authors suggest a simple way to generate phase singularities using photochemical reactions. This resulted in all-optical control of resonator-free singular phase points that could have applications in biosensing, for example. The presentation of results is clear and I suggest to publish the manuscript after addressing the following points.

1. Page 2 "We use photochromic molecules to control the number of molecules, and hence light-matter coupling strength, by simple irradiation of light[19]". This sentence is dubious and needs to be amended (which number of which molecules do you control?).

2. Page 2 "Our results demonstrate what we believe to be the first all-optical control of phase singularities and the first application of strong light-matter coupling to the creation of phase singularities in a remarkably simple structure."

I am not really sure about this statement as one can easily get phase singularities in an interferometer which can be also controlled by light - there was a lot of activity in this field.

3. The thickness dependence on many Figures appears to be "digital". This is probably connected with the fact that films of some thicknesses were studied. If this is the case, why the results for these films are shown as a line (not a point) on many Figures (say Figure 1)?

4. Discussion section somehow appears to be a conclusion section. Does it mean that there is nothing to discuss? Then, it would be nice to finish with just Conclusion section. If there is something to discuss, it would be nice to have a proper discussion section.

Dear Editor,

We thank you for considering our work for publication in the journal *Nature Communications*. We have carefully read the thoughtful comments made by all the Reviewers. We have now addressed all of their comments, resulting in substantial improvements to the manuscript.

All the changes we have made are highlighted in the revised manuscript. Modifications made in response to the Reviewers' comments are listed below.

Kind regards,
Philip Thomas
On behalf of all authors

Reviewer #2

Manuscript "All-optical control of phase singularities using strong light-matter coupling" by P. A. Thomas, K. S. Menghrajani and W. L. Barnes is an experimental work demonstrating the appearance of "phase singularities" in a new relatively simple cavity-free setting. Authors utilize strong coupling of laser light to a thin film of photochromic molecules and formation of polaritons. Controlling the strength of the coupling by the number of resonant molecules, authors demonstrate the appearance of anticrossings in the polariton branches of the energy vs. film thickness dispersion in Fig. 1. Corresponding phase profiles in Fig. 3 confirm appearance of dislocations with opposite twists. The work is technically sound and represent sufficient novelty of interest to the broad audience of Nature Communications.

We thank the Reviewer for taking the time to review our manuscript and are grateful for their comments. We have used the Reviewer's feedback to revise our manuscript, we think it is stronger as a result.

My main concern is what appears to be a confusing terminology emerging in related fields, namely use of "phase singularities" outside of the original context introduced by J. F. Nye and M. V. Berry in 1974 [Dislocations in Wave Trains, Proc. R. Soc. London A 336, 165 (1974)]. The later wavefront phase singularities, or optical vortices, as field objects are closely related in their features to "phase dislocations" observed in current manuscript and related works, e.g. quantization of the twist around dislocations, nucleation of vortex-antivortex pairs, etc. Despite their differences they indeed must belong to a general class of optical phenomena reviewed recently by J. Ni et al as "Multidimensional phase singularities in nanophotonics" [Science 374, 418 (2021)], distinguishing singularities in the spatial, spatiotemporal, and momentum domains.

Similarly, the term "singular phase optics" is in direct confusion with "singular optics" of transverse laser modes carrying optical angular momentum, see, e.g.

M. S. Soskin and M. V. Vasnetsov, Prog. Opt. 42, 219 (2001),

A. S. Desyatnikov, Yu. S. Kivshar, and L. Torner, Prog. Opt. 47, 291 (2005),

M. R. Dennis, K. O'Holleran, and M. J. Padgett, Prog. Opt. 53, 293 (2009).
Ironically, the only mentioning of “phase singularity” in Ref. [1] of present manuscript is related to wavefront singularity.

Authors are encouraged to put their findings in a broad context of established field of singular optics and references above, in particular discuss how “phase singularities” in the polariton branches of measured spectra here may relate to the spatiotemporal wave singularities of polaritons and their light component.

We thank the Reviewer for raising important points concerning terminology. We have clarified or removed unhelpful references to “singular phase optics” in the Abstract, Introduction and Conclusion. We have also rewritten the opening paragraph of the Introduction in a way which we hope better places our work in the context of the established field. We have also expanded our Discussion, highlighting the parallels between our phase singularities and those observed in the spatiotemporal domain.

Reviewer #3

It is quite difficult to understand the purport of this paper. The experimental results which demonstrate the possibility to create phase singularities in optical fields hosted by thin field of molecular materials may be, in principle, appropriate for the publication. However, the publication in Nature Communications implies that the paper should be sufficiently interesting to broader readership of the journal, being clearly related to sufficiently general and clearly defined topics. The present manuscript does not meet this condition. The presentation is strongly focused on details of the experiment, without showing an intention to relate the results to general themes of optics.

We thank the Reviewer for taking the time to examine our manuscript and are grateful that they regard our findings as publishable. We have used the Reviewer's feedback to make improvements to our manuscript.

In particular, a commonly known fact is that phase singularities appear in cores of optical vortices. The present manuscript completely ignores this relation. Such concepts as vortices or vorticity are not even mentioned in the text.

We thank the Reviewer for this comment and agree that our original introduction of phase singularities was too brief. We have rewritten the opening paragraph to better introduce the concepts of phase singularities, singular optics and the difference between phase singularities associated with optical vortices and the phase singularities we discuss in the present manuscript. We have also discussed optical vortices in our expanded Discussion section.

Further, it is commonly known that phase singularities are characterized by the topological charge (winding number). If it is S , then the amplitude field around the singularity decays at

$r \rightarrow 0$ as r^S (r is the distance from the similarity point). The text twice quickly mentions the topological charge, without specifying values of the charge observed in the experimental pictures,

We thank the Reviewer for raising this point. We have revised the manuscript to more quantitatively describe the topological charges observed in our results.

and no result are reported or even briefly mentioned for the amplitude structure, that should be related to the phase singularities.

We thank the Reviewer for making this comment. In response we have added an additional Supplementary Figure, which shows the amplitude counterparts to the experimental and calculated phase dispersion plots in Fig. 3. We think that this allows for a better side-by-side comparison of amplitude and phase data than the amplitude dispersion plots in Fig. 1.

A paper which is so tightly focused on technicalities of observing phase distributions of optical fields in thin molecular films, without placing it in the general context of optics and physics of vortices, may be appropriate for a strongly specialized journal, such as Applied Optics or, maybe, Optics Express. The paper, in its present form, seems completely inappropriate for an interdisciplinary journal, such as Nature communications.

We thank the Reviewer for encouraging us to broaden the focus of our manuscript. In our Introduction we have sought to emphasise the relevance of our findings to the fields of biosensing and optoelectronics. We have also expanded both our Introduction and Discussion sections to better place our findings in the context of singular optics.

Reviewer #4

Authors suggest a simple way to generate phase singularities using photochemical reactions. This resulted in all-optical control of resonator-free singular phase points that could have applications in biosensing, for example. The presentation of results is clear and I suggest to publish the manuscript after addressing the following points.

We thank the Reviewer for taking the time to examine our manuscript and are grateful for their comments below which have allowed us to improve the manuscript.

1. Page 2 "We use photochromic molecules to control the number of molecules, and hence light-matter coupling strength, by simple irradiation of light[19]". This sentence is dubious and needs to be amended (which number of which molecules do you control?).

We thank the Reviewer for pointing out the ambiguity in this sentence. We have rewritten it as follows: "We use a photochemical reaction to control the number of

coupled molecules, and hence light-matter coupling strength, by simple irradiation of light."

2. Page 2 "Our results demonstrate what we believe to be the first all-optical control of phase singularities and the first application of strong light-matter coupling to the creation of phase singularities in a remarkably simple structure."

I am not really sure about this statement as one can easily get phase singularities in an interferometer which can be also controlled by light - there was a lot of activity in this field.

We thank the Reviewer for pointing out this error. We have removed the phrase "the first all-optical control of phase singularities" from the manuscript.

3. The thickness dependence on many Figures appears to be "digital". This is probably connected with the fact that films of some thicknesses were studied. If this is the case, why the results for these films are shown as a line (not a point) on many Figures (say Figure 1)?

We thank the Reviewer for this comment. The Reviewer is correct that we could re-plot these results as lines instead of colour plots. However, we feel that the use of line plots would make it harder for us to identify pairs of phase singularities and their topological charges, we have thus left the figures in the same form as originally submitted.

4. Discussion section somehow appears to be a conclusion section. Does it mean that there is nothing to discuss? Then, it would be nice to finish with just Conclusion section. If there is something to discuss, it would be nice to have a proper discussion section.

We thank the Reviewer for suggesting that we expand our Discussion section. We have added an extra paragraph to the Discussion section in which we compare our findings to existing results in the literature.

REVIEWERS' COMMENTS

Reviewer #2 (Remarks to the Author):

Authors fully addressed my concerns and revised the manuscript accordingly; it can be published as it is now.

Reviewer #4 (Remarks to the Author):

The authors answered all my points and I am happy to recommend the manuscript for publication.

Reviewer #2 (Remarks to the Author):

Authors fully addressed my concerns and revised the manuscript accordingly; it can be published as it is now.

We thank the reviewer for their support of our work.

Reviewer #4 (Remarks to the Author):

The authors answered all my points and I am happy to recommend the manuscript for publication.

We are grateful for the reviewer's support of our work.